# Genomic characterization of a novel, widely distributed *Mycoplasma* species "*Candidatus* Mycoplasma mahonii" associated with the brittlestar *Gorgonocephalus chilensis*

Oluchi Aroh[1]*, Mark R. Liles[1], Kenneth M. Halanych[1,2]*

1 Department of Biological Sciences, Auburn University, Auburn, AL, United States of America, 2 Centre for Marine Science, University of North Carolina Wilmington, Wilmington, NC, United States of America

* olo0002@auburn.edu (OA); halanychk@uncw.edu (KMH)

**Data Availability Statement:** All relevant data are within the paper and its Supporting Information files.

## Abstract

Symbiotic relationships are ubiquitous throughout the world's oceans, yet for many marine organisms, including those in the high latitudes, little is understood about symbiotic associations and functional relationships. From a recently determined genome sequence of a filter-feeding basket star from Argentina, *Gorgonocephalus chilensis*, we discovered a novel *Mycoplasma* species with a 796Kb genome (CheckM completeness of 97.9%, G+C content = 30.1%). Similar to other *Mycoplasma* spp. within Mycoplasmatota, genomic analysis of the novel organism revealed reduced metabolic pathways including incomplete biosynthetic pathways, suggesting an obligate association with their basket star host. Results of 16S rRNA and multi-locus phylogenetic analyses revealed that this organism belonged to a recently characterized non-free-living lineage of *Mycoplasma* spp. specifically associated with marine invertebrate animals. Thus, the name "*Candidatus* Mycoplasma mahonii" is proposed for this novel species. Based on 16S rRNA PCR-screening, we found that *Ca*. M. mahonii also occurs in *Gorgonocephalus eucnemis* from the Northwest Pacific and other *Gorgonocephalus chilensis* from Argentinian waters. The level of sequence conservation within *Ca*. M. mahonii is considerable between widely disparate high-latitude *Gorgonocephalus* species, suggesting that oceanic dispersal of this microbe may be greater than excepted.

## Introduction

*Mycoplasma* species are one of the smallest and simplest self-replicating organisms, with a very reduced genome size that can range from about 540 to 1300 Kb. Mycoplasmas possess the minimum set of genes essential for growth and replication and evolved from the *Bacillus/Clostridium* branch of Gram-positive eubacteria by reductive evolution [1, 2]. These bacteria lack a cell wall, a feature responsible for their pleomorphism and their resistance to some antibiotics [2].

**Funding:** 1. KMH - OPP-1916661 - National Science Foundation- https://www.nsf.gov/ 2. KMH - OCE-1155188 - National Science Foundation- https://www.nsf.gov/ 3.KMH - ANT-1043745 - National Science Foundation- https://www.nsf.gov/ 4. KMH - OPP-0338218 - National Science Foundation- https://www.nsf.gov/ The funders had no role in study design, data collection and analysis, decision to publish, or preparation of the manuscript.

**Competing interests:** The authors have declared that no competing interests exist.

Mycoplasmas encompass over 100 species and usually live in close association with their plant or animal host(s) to fulfill nutritional requirements [3]. Pathogenic mycoplasmas are responsible for numerous respiratory infections in humans such as pneumonia, as well as other conditions including pelvic inflammatory disease and urethritis [4]. In addition, the existence of mycoplasmas has been successfully documented in marine organisms including fishes (where they mainly colonize the intestines, gills, liver, and kidney) [2, 5, 6], cold-water corals [7], lobster [8], octopus [9, 10], abalone [11, 12] and squid [10]. They have also been found to be associated with the microbiota of echinoderms such as the blue bat star *Patiria Pectnifera* [13]. In these studies, mycoplasmas were in a commensal association with their host organism except in salmonids [2] where a mutualistic association was observed.

In this study, we report the discovery of a *Mycoplasma* with a 796kb genome (CheckM completeness of 97.9%) in the tissue of *Gorgonocephalus chilensis*, a filter-feeding basket star. To promote understanding of *Mycoplasma* spp. diversity, and symbiosis with marine invertebrates, we explored the new species' genomic composition and inferred metabolic capabilities. Additionally, given the understudied environment from which this cold-water filter-feeding echinoderm was discovered, we explored the degree of relatedness between *Ca*. M. mahonii and other *Mycoplasma* spp. using 16S rRNA and multilocus phylogenetic analyses. and whether this novel species occurred in other basket stars. To the best of our knowledge, this is the first genetic isolation and characterization of mycoplasmas in the *Gorgonocephalus* genus and the name *Candidatus* Mycoplasma mahonii is proposed for this novel species.

## Materials and methods

### Genome sequencing

Genomic DNA of *Gorgonocephalus chilensis* (Ophiuroidea, Euryalida, Gorgonocephalidae) was extracted from an individual sampled in Argentinian waters during an *ARSV Laurence M. Gould* expedition (LMG-0605, May 2006, latitude; 54˚49, longitude -60˚16, depth 110m). Tissue was stored at -80˚C and subsequently sent to the University of Arizona Genomics Institute where DNA was isolated and sequenced using PacBio CCS long read technology using the protocol outlined below.

High molecular weight (HMW) DNA was extracted from ground tissue in extraction buffer with Tris HCl buffer 0.1M pH 8.0, EDTA 0.1M pH8, SDS 1%, and Proteinase K in 50˚C for 30 minutes. The mixture was spun down and the aqueous phase transferred to a new tube. Next, 5M Potassium acetate was added, precipitated on ice, and spun down. After centrifugation, the supernatant was gently extracted with 24:1 chloroform: isoamyl alcohol. The upper phase was transferred to a new tube and DNA precipitated with isopropanol. DNA was collected by centrifugation, washed with 70% ethanol, air dried, and dissolved thoroughly in 1x TE followed by RNAse treatment. The DNA purity was measured using Nanodrop, DNA concentration was measured with Qubit HS kit (Invitrogen), and DNA size was validated by the Femto Pulse System (Agilent). The extracted HMW DNA was sheared to appropriate size range (10–30 kb) using Megaruptor 3 (Diagenode). The sequencing library was constructed following manufacturers protocols using SMRTbell Express Template Prep kit 2.0. The final library was size selected on a Blue Pippin (Sage Science) using S1 marker with a 10–25 kb size selection. The recovered final library was quantified with Qubit HS kit (Invitrogen) and size checked on Femto Pulse System (Agilent). The sequencing library was prepared with PacBio Sequel II Sequencing kit 2.0 for HiFi library, loaded to 8M SMRT cells, and sequenced in CCS mode in the Sequel II instrument for 30 hours.

Raw reads were assembled using HiFiasm [14] (N50 = 1057833bp, Genome size = 3.5GB). The assembled genome of *G. chilensis* was screened for bacterial 16S rRNA by BLASTn against

the NCBI's 16S rRNA database [15]. Contigs that matched to 16S rRNA genes were assigned taxonomic labels using Kraken2 Silva database v.138 [16], with default parameters, and were further analyzed. For all software used herein, default parameters were employed unless otherwise noted.

Completeness of each matched contig was determined with CheckM [17], using default parameters. Contigs having completeness higher than 90% and contamination lower than 10% were considered a "complete" metagenomic assembly. Taxonomic assignments of complete contigs were conducted using GTDB-Tk v1.6.0 [18], based on the Genome taxonomy database [19]. GTDB-Tk uses a combination of metrics, including average nucleotide identity to reference genomes in the NCBI Assembly database, placement in the GTDB reference tree, and the relative evolutionary divergence.

## Genome annotation of novel *Mycoplasma*

For the complete bacterial genome, gene and subsystem annotation was conducted using RAST v2.0 [20]. RNAs were annotated using RNAmmer v1.2 [21]. The Kyoto Encyclopedia of Genes and Genomes (KEGG- BlastKOALA) v2.2 [22] was used to predict biological pathways present in the genome. Ori-Finder 1 [23, 24] was used to identify the Origin of Replication (OriC). Clusters of orthologous genes (COGs) were annotated using eggnog-mapper v2.1.7 [25]. The genome was scanned for virulence factors using the BLAST search tool of the Virulence Factor Database (VFDB) [26] (database used–core and full dataset protein sequence). CRISPRCasFinder v1.1.2 [27] was used to validate predicted CRISPR/CAS systems. The average nucleotide identity (ANI) value was calculated using the Ezbio ANI calculator tool [28], and the average amino acid identity (AAI) was calculated using the webserver available through the Georgia Institute of Technology [29]. The dDDH calculator from Type Strain Genome Server (TYGS) [30] was used to predict digital DNA:DNA hybridization (dDDH) values from intergenomic distances for *Ca*. M. mahonii and its most closely related type strains genome sequences as implied in the Genome to Genome distance calculator (GGDC) [31].

## Screening for *Mycoplasma* from multiple *Gorgonocephalus* individuals

Fifteen *Gorgonocephalus* individuals were screened for *Mycoplasma*. Nine *G. chilensis* from Argentinian waters were obtained during the LMG-0605 cruise in 2006 and 6 *G. eucnemis* were obtained in 2014 near the University of Washington's Friday Harbor laboratories in the Northeast Pacific; S1 Table. All samples were collected by KMH by trawl and subsequently stored in a -80°C freezer. Total genomic DNA was extracted using the Qiagen DNeasy blood and tissue kit (Maryland, USA) following the manufacturer's instructions except that lysis was done overnight to ensure complete digestion due to the calcium carbonate in the arm tissue. Agarose gel electrophoresis was used to verify the integrity of the isolated DNA.

A 716bp region of the 16S RNA gene was targeted for PCR amplification using two oligonucleotide primers (Forward- 5′-ACTCCTACGGGAGGCAGCAGTA-3′; Reverse 5′-TGCAC CATCTGTCAYTCYGTTAACCTC-3′) that were slightly modified from previously published *Mycoplasma* universal oligopeptide primers [32] to be more specific to *Ca*. M. mahonii. Thermocycling conditions included an initial denaturation at 98°C for 30 sec, followed by 35 cycles of denaturation at 98°C for 10 sec, annealing at 60°C– 62°C for 30 sec, extension at 72°C 30 sec, and a final extension at 72°C for 5 min. Negative controls were employed in PCRs, samples from Argentinian waters and Northeast Pacific samples were handled in the lab on different dates. Amplified PCR products were examined by 1% gel electrophoresis and purified using the Qiagen QIAquick purification kit (Maryland, USA). The purified template was

Sanger sequenced by Genewiz (New Jersey, USA), and bidirectional reads were trimmed and verified using Geneious software (v 2021.2.2) [33].

## Phylogenetic analysis

To determine the phylogenetic affiliation of the new organism, two sets of phylogenetic analyses were conducted. One analysis employed a single gene tree based on 16S rRNA data, allowing for greater taxon sampling. The second analysis employed a multilocus tree, albeit with fewer taxa, to overcome the bias that can be caused by single-gene analysis (e.g. lineage sorting and selection pressure) and to provide a more robust representation of genomic data.

All available 16S rRNA sequences from *Mycoplasma* type strain were obtained from Ezbiocloud database [34] and GenBank [35], additionally, Mycoplasmatales bacteria DT_67 and DT_68, as well as 2 Oyster Mollicutes MAGs, were added to sequences obtained from this study to reconstruct the phylogenetic history of the group. *Bacillus subtilis* was used as an outgroup to root the resultant trees based on current understanding of *Mycoplasma* evolutionary relationships [36, 37]. Sequences were aligned using MAFFT v7.475 [38], with default parameters. Maximum likelihood analysis with bootstrap was employed to reconstruct phylogenetic relationships in IQtree v1.6.12 [39] using the following parameters '-bb 100000, -nt AUTO,—runs 5'. These parameters were employed for all IQtree phylogenetic analyses in this study. The substitution model used for the 16S rRNA gene-based phylogeny, GTR+F+R10, was selected as the best model by IQtree's ModelFinder.

For the multilocus phylogenetic analysis, a representative with a sequenced genome from each major clade present in the 16S rRNA tree was selected and its genome screened for the presence of five single-copy housekeeping genes–*recA*, *lepA*, *dnaK*, *ruvB*, and *gmk*. This gene choice was based on the available literature for *Mycoplasma* phylogeny [40–44]. DNA sequences of these 5 housekeeping genes were aligned using the MAFFT option in Geneious (Geneious Prime 2023.0.4, Java version 11.0.15+10, 64-bit) and then concatenated. The multilocus phylogenetic tree of the concatenated alignment was reconstructed by employing maximum likelihood analysis with bootstrap in IQtree v1.6.12. The substitution model used for the multilocus phylogeny, GTR+F+R6, was selected as the best model by IQtree's ModelFinder. Additionally, tree topologies for the individual genes were also inferred using a MAFFT v7.475 alignment and maximum likelihood in IQtree v1.6.12. A Bayesian Inference analysis was also run for both the 16S rRNA and multilocus tree using MrBayes [45]. The same set of sequences as mentioned earlier was used, and the GTR+I+G model was chosen through JModeLTest2 [46]. on CRIPES Science Gateway (https://www.phylo.org/index.php/ (accessed on April 2023)), and with 1000000 generations sampled every 500 generations. Burninfrac was set to 0.25. All phylogenetic trees were visualized and edited using Figtree v1.4.4 [47].

## Ethics statement

All fieldwork was carried out under the auspices of the USA Antarctic program or the University of Washington's Friday Harbor Laboratories, with all applicable rules and permitting requirements followed.

## Results

### Microbial identification and classification

BLAST results of the *G. chilensis* assembly against NCBI 16S rRNA genes revealed 4 contigs that contained bacterial 16S rRNA gene fragments. Three contigs were classified as *Mycoplasma* while one was unclassified by the Kraken2 silva database. The 3 classified contigs were

51 Kb, 73 Kb, and 796 Kb in size respectively, 16S rRNA gene percentage identities of the 769kb contig compared to the other contigs ranged from 75% - 88%. Previously described *Mycoplasma* spp. genome sizes range from 540 Kb to 1300 Kb, hence we hypothesized that the 51 Kb and 73 Kb contig were likely to be incomplete or fragments of *Mycoplasma* genomes. CheckM analyses confirmed this interpretation as completeness of 5.72%, 7.14%, and 97.93% were reported for the 73 Kb, 51 Kb, and 796 Kb contigs respectively. In addition, CheckM only reported 0.38% contamination for the 796 Kb contig. Hence, we considered the 796 Kb contig to represent a complete genome, and further downstream analyses were conducted only on this MAG.

GTDB-Tk robustly placed this complete MAG in the order Mycoplasmatales and family Metamycoplasmataceae (GTDB-tk RED value of 0.93; S2 Table) and phylogenetic analysis shows it to be related to a marine clade of *Mycoplasma*. This identified novel *Mycoplasma* genome was designated as "*Candidatus* Mycoplasma mahonii" (formal description given below).

## Genome annotation

**General features of the genome.** The novel *Ca*. M. mahonii genome consists of a single chromosome of 796,768 bp with a GC content of 30.1% (Fig 1). The 16S, 23S, and 5S rRNA genes were present as single copies, with the 16S and 23S rRNA genes located in the same operon and the 5S rRNA gene in a separate genomic region. Thirty-one transfer RNAs (tRNA) were identified, and all standard amino acids were represented. RAST predicted a total of 780 protein-coding sequences (CDS) of which 406 CDS (52.1%) were assigned putative functions and 374 CDS (47.9%) were annotated as hypothetical proteins. Repeats comprised 6.9% of the genome. Average gene length of predicted CDS was 887bp. Among the predicted CDS, 397 CDS (50.8%) were classified into Clusters of Orthologous (COG) families comprising 18 functional categories with most genes belonging to the J class (Translational, ribosomal structure, and biogenesis). RAST and eggNOG (COG) annotation were similar to that seen in other *Mycoplasma* spp. genomes based on RAST subsystems (Fig 2 and S3 Table) and COG classifications (Fig 3 and S4 Table). Additionally, RAST subsystem category gene counts were compared between *Ca*. M. mahonii and closely related species (S1 Fig).

To verify the functional abilities of the recovered MAG (i.e., the proposed genome), we examined a series of molecular and cellular pathways and structures, and below we describe select major systems in turn.

**Replication, transcription, and translation.** For DNA replication in *Mycoplasma*, the OriC usually contains *dnaA* boxes and is generally around the vicinity of the *dnaA* gene [48–51]. Using previously published *dnaA* box motif consensus sequence 5'-TTATCCACA-3' [48, 52], and allowing one mismatch, two putative replication origins were found in the area surrounding *dnaA* gene using Ori-Finder. Both regions possessed the typical features of OriC in prokaryotes (i.e. *dnaA* box, *dnaA* gene vicinity, GC skew inversion) [53].

A total of 43 genes (S5 Table) were predicted to be involved in replication, recombination, and repair. DNA repair appears to be mainly executed by nucleotide excision repair, SOS repair system, and recombination repair. No mismatch-repair system genes (MutHLS) were found.

For transcription, a total of 11 genes were predicted to be involved (S6 Table). Transcription termination and elongation are regulated by *Nus A*, *B*, *G*, and *GreA* genes in this organism. *GreA* prevents transcription arrest while the Nus proteins can induce transcription pausing or stimulate anti-termination [54]. In addition, two transcription factors (*HigA* and *MraZ*) and one heat-inducible transcription repressor gene (*HrcA*) were identified.

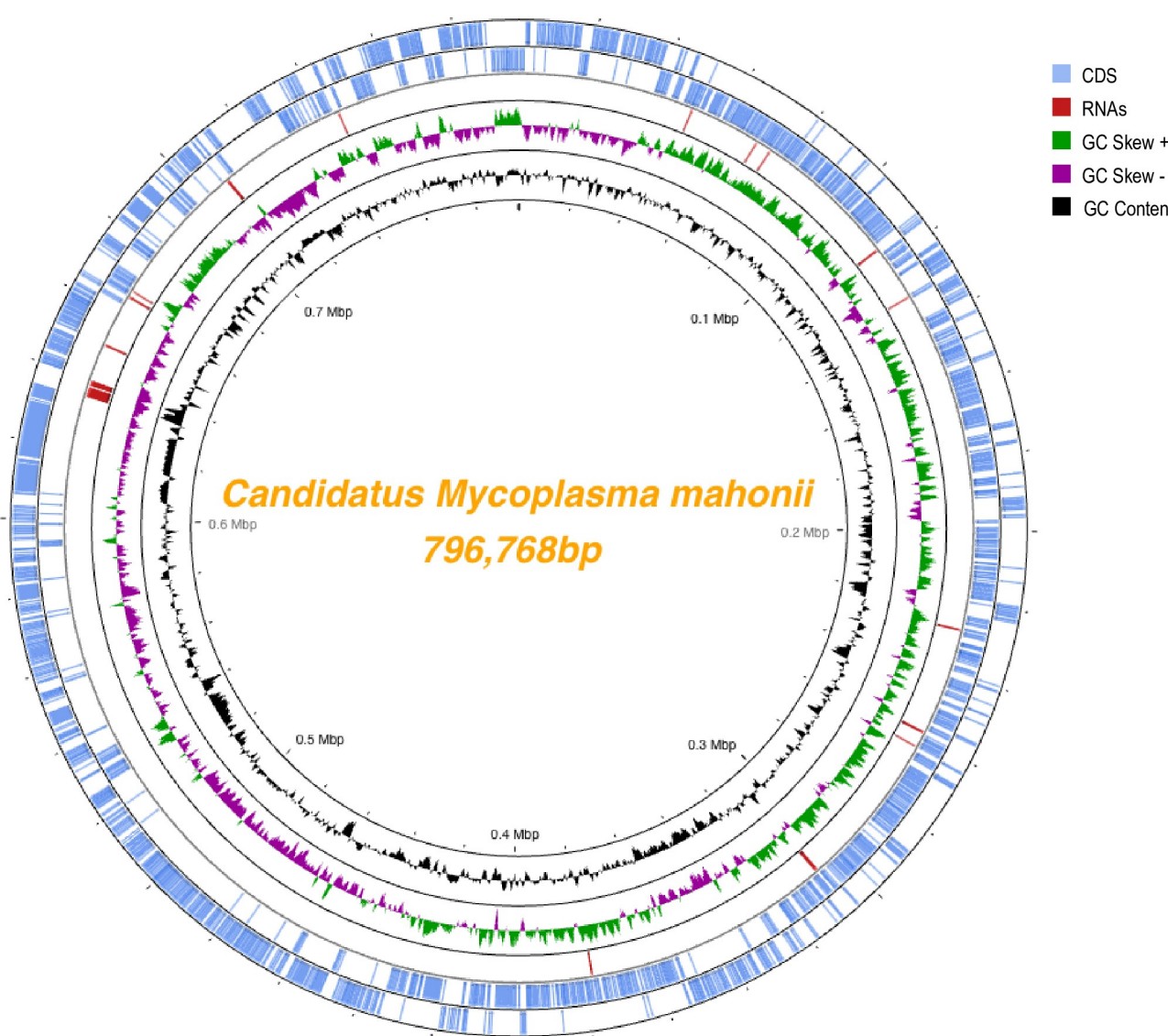

**Fig 1. Chromosome atlas of *Candidatus* Mycoplasma mahonii.** The scale is shown by the inner black circle. Starting with the outermost rings, the 1st and 2nd circles show predicted coding sequences on the minus and plus strands respectively. The 3rd circle represents RNAs including both tRNAs and rRNAs. The 4th circle represents GC skew (G−C)/(G+C) (green-above mean, purple-below mean; mean = 0.5) and the 5th circle represents mean-centered G+C content of the genome. The figure was generated using Proksee.

Additionally, a total of 126 genes were predicted to be involved in translation, ribosomal structure, and biogenesis including 49 ribosomal proteins, 24 aminoacyl tRNA synthase genes, and 11 translation factors (S7 Table).

**Secretion system and transporters.** The transporter system of *Ca*. M. mahonii consists of 39 genes, which are mainly made up of the ATP-binding cassette (ABC) transporter system and the phosphotransferase (PTS) system (S8 Table). For the PTS system, 4 genes that encode proteins required by the PTS system were present; *pts1* which encodes Enzyme 1 (E1), *ptsH* which encodes Phosphocarrier protein Hpr, *fruA* which encodes a fructose-specific II component (EIIBC or EIIC) and *celA* which encodes a cellobiose-specific II component (EIIB).

Genes encoded by the ABC transport system include 12 ATP-binding proteins, 11 permease proteins, and 2 substrate-binding proteins. Three complete ABC-type transport systems

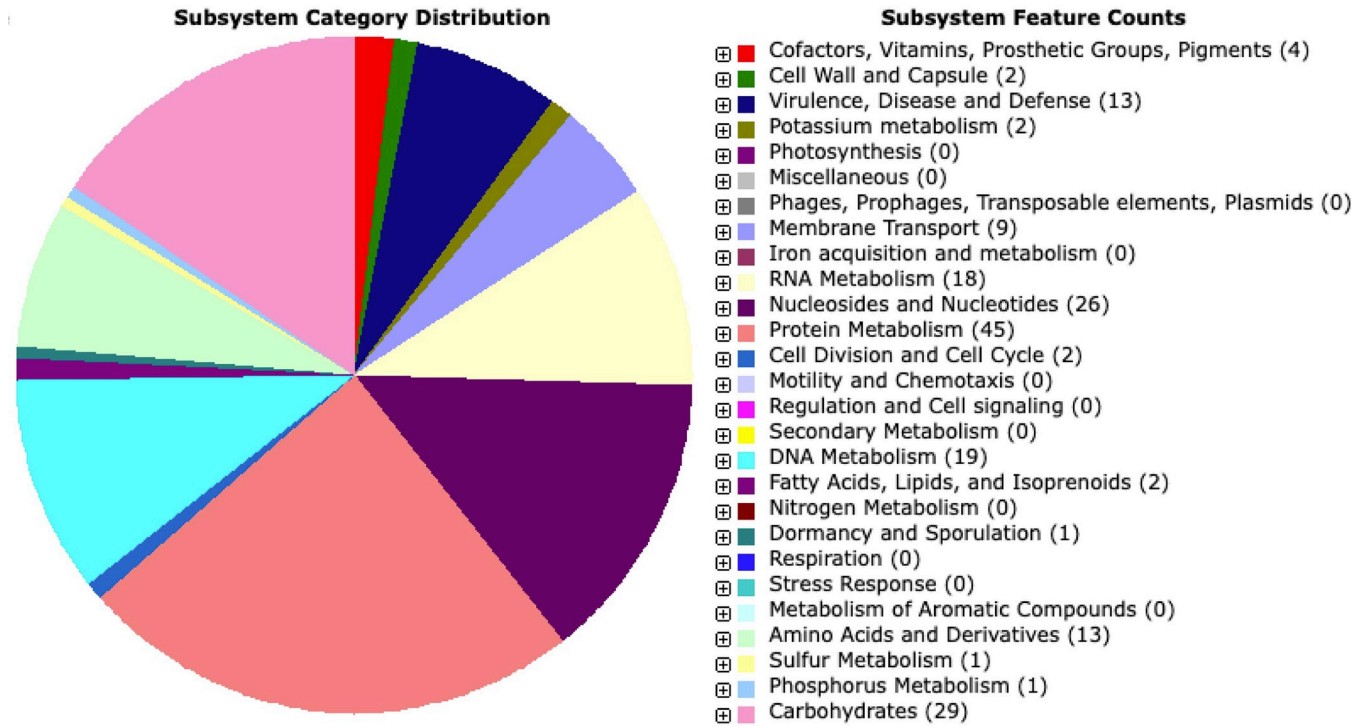

**Subsystem Category Distribution**

**Subsystem Feature Counts**

- Cofactors, Vitamins, Prosthetic Groups, Pigments (4)
- Cell Wall and Capsule (2)
- Virulence, Disease and Defense (13)
- Potassium metabolism (2)
- Photosynthesis (0)
- Miscellaneous (0)
- Phages, Prophages, Transposable elements, Plasmids (0)
- Membrane Transport (9)
- Iron acquisition and metabolism (0)
- RNA Metabolism (18)
- Nucleosides and Nucleotides (26)
- Protein Metabolism (45)
- Cell Division and Cell Cycle (2)
- Motility and Chemotaxis (0)
- Regulation and Cell signaling (0)
- Secondary Metabolism (0)
- DNA Metabolism (19)
- Fatty Acids, Lipids, and Isoprenoids (2)
- Nitrogen Metabolism (0)
- Dormancy and Sporulation (1)
- Respiration (0)
- Stress Response (0)
- Metabolism of Aromatic Compounds (0)
- Amino Acids and Derivatives (13)
- Sulfur Metabolism (1)
- Phosphorus Metabolism (1)
- Carbohydrates (29)

**Fig 2. RAST subsystem feature categories.** The chart is color-coded based on the corresponding subsystem category.

including a phosphate transporter, a phosphonate transporter, and a general nucleoside transporter were present, while others such as oligopeptide transporter, energy-coupling factor transporter, saccharide/lipid transporter, etc. were incomplete. Other genes that were not part of the PTS system or the ABC transport system were associated with other transporters such as magnesium transporter, riboflavin transporter, potassium uptake system proteins, cytosine permease, and an adenine/guanine/hypoxanthine permease.

Genes such as *secA*, *secY*, *secD/F*, *secE*, *secG*, *ffh*, *yidC*, and *FtsY*, which make up the core proteins required for the Sec-SRP secretion pathway (SEC system) and two other genes associated with the Type II secretion system (*comEB* and *comEC*) were present as part of the organism's bacterial secretion system (S9 Table).

**Metabolism.** According to the annotation data, *Ca*. M. mahonii encode all needed enzymes in the Embden-Meyerhof-Parnas (EMP or glycolysis) pathway, arginine deaminase pathway, F1-ATP synthase, PRPP (phosphoribosyl-pyrophosphate) production, and pyruvate oxidation. Enzymes associated with lipid metabolism in this organism suggest that they utilize glycerol to generate phospholipids (cardiolipin), the only membrane component identified in this organism.

Genes involved in de novo nucleotide synthesis were lacking in *Ca*. M. mahonii; however, this bacterium was predicted to encode nucleotide salvage pathways genes such as *nrdA* and CTP synthase. *Ca*. M. mahonii lacks genes predicted to be required for different intermediate pathways such as the TCA cycle, citric acid cycle and the oxidative phase of the pentose phosphate pathway. Partial pathways for CoA biosynthesis (specifically pantothenate to CoA), phospholipid biosynthesis, riboflavin/FMN/FAD biosynthesis, Tetrahydrofolate (THF) biosynthesis amongst others were present within the *Ca*. M. mahonii genome.

**Defense system.** Two bacterial defense systems to ward off phage infection, including a Type II Restriction—modification (R-M) system and a CRISPR-CAS system are present in the

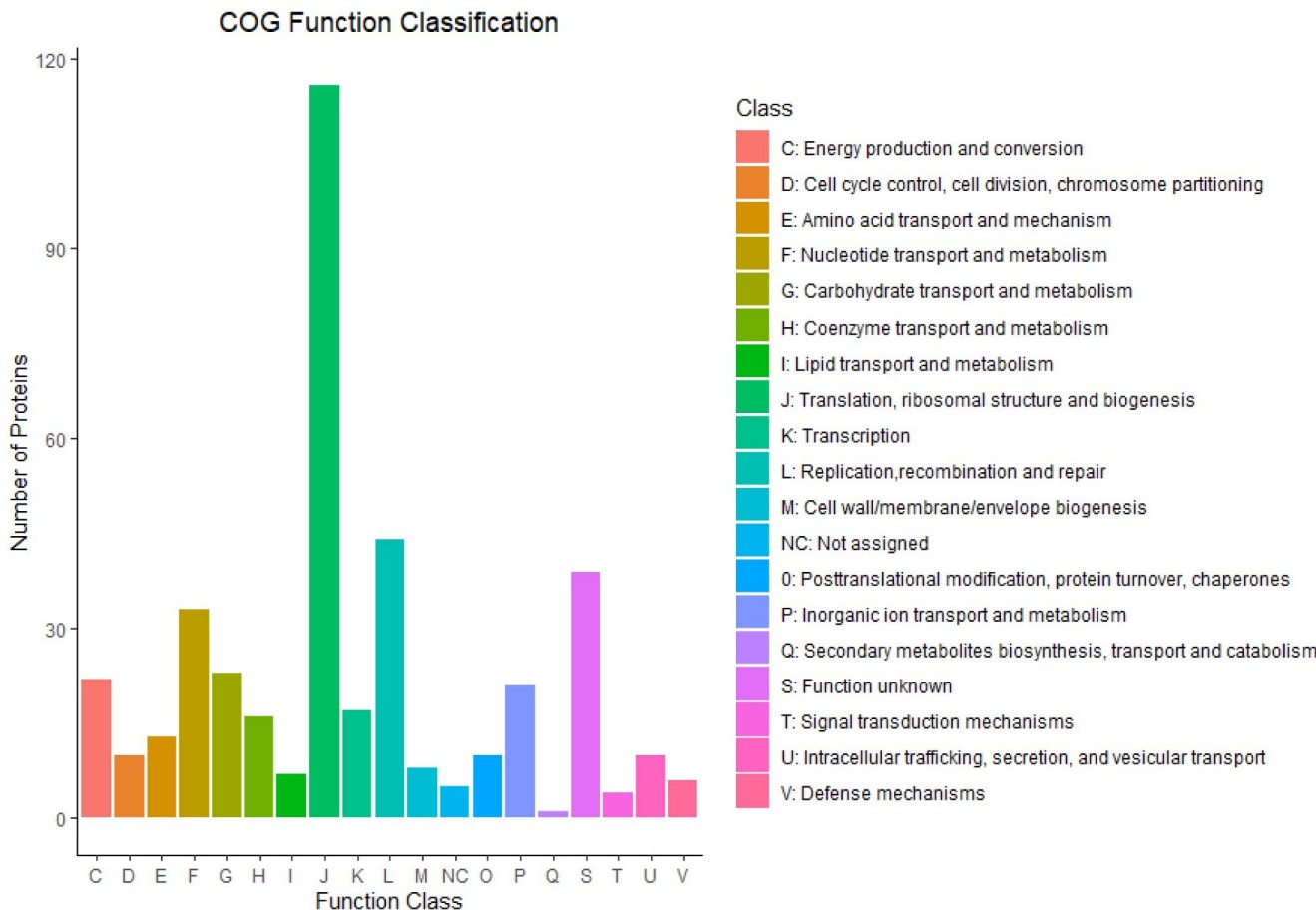

**Fig 3. Clusters of orthologous group function classification.** The Y-axis represents the number of protein/unigenes belonging to a particular category while the X-axis represents the COG categories.

genome of *Ca*. M. mahonii. The Type II restriction system possessed only genes encoding methylase and methyltransferase enzymes but lacks the sequence-specific endonuclease which is usually responsible for recognizing and cleaving specific DNA sequences. No Type I or Type III R-M genes were found. On the other hand, RAST annotation (and CRISPR finder tools) identified 1 CRISPR array of length 9820bp, 148 spacers, and the direct repeat consensus GTTTAAGAATACACAAGAATGATACCACCCCAAAAC. Additionally, a thioredoxin reductase system which is predicted to provide defense against oxidative stress was also present.

## Screening for Mycoplasmas in *Gorgonocephalus*

Nine out of 15 screened basket star samples (6 *G. chilensis* samples from Argentinian waters and 3 *G. eucnemis* samples from the Northeast Pacific) had a positive PCR result and these amplicons were Sanger sequenced. However, only 1 *G. eucnemis* sample had a good quality read and was the only *G. eucnemis* sample used in the phylogenetic analysis. These sequences have been deposited to NCBI under the accession numbers OP995472-OP995479.

## Phylogenetic results

16S rRNA gene-based (Fig 4) and multilocus (Fig 5) phylogenetic analyses were used to validate the GTDB-Tk taxonomical assignment and provide higher resolution phylogenetic

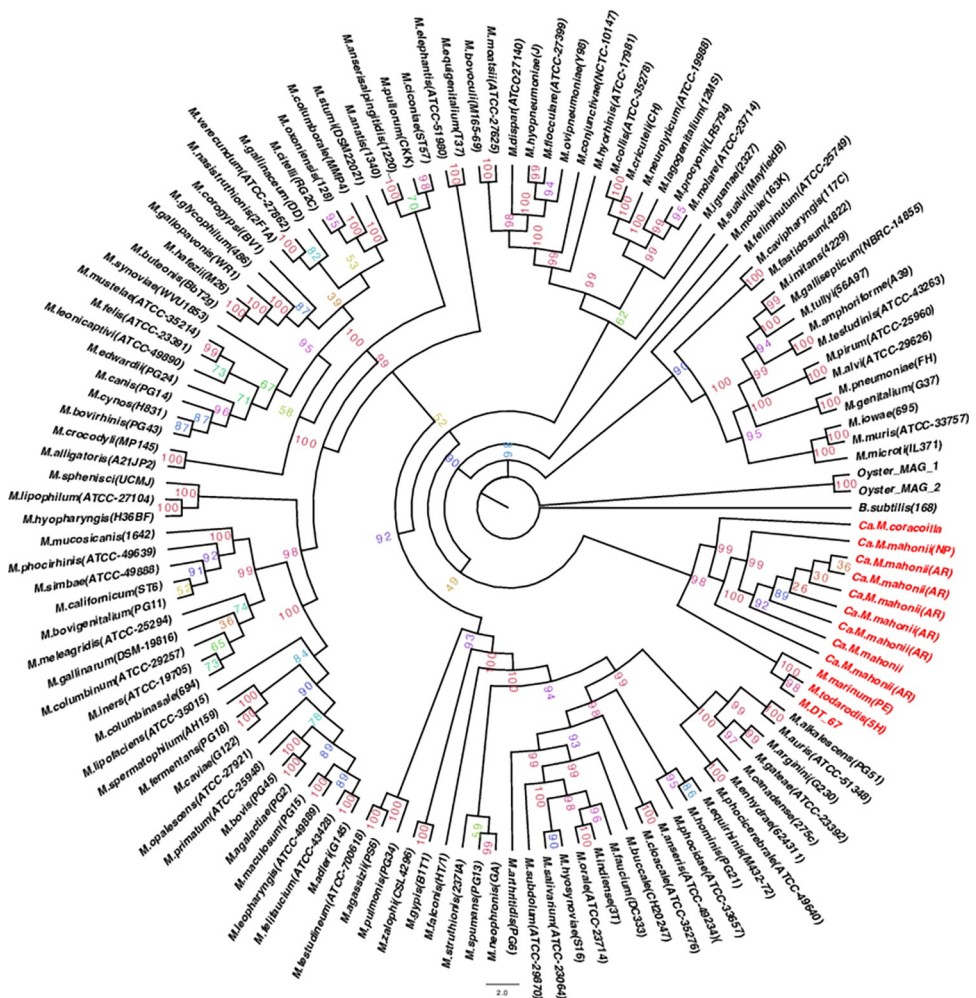

**Fig 4. 16S rRNA maximum-likelihood phylogenetic tree.** The 16S rRNA gene tree was reconstructed based on all available *Mycoplasma* spp. type strain available from the Ezbiocloud and GenBank databases, Mycoplasmatales bacteria DT_67 and DT_68, 2 Oyster Mollicutes MAG and *Ca.* M. mahonii. Bootstrap percentage values are shown on the tree. The tree was generated in IQtree with the GTR+F+R10 model. *Ca.* M. mahonii and other sequences making up the distinct marine clade are shaded red. AR–Argentinian waters samples, NP–North Pacific samples.

placement. The 16S rRNA gene analysis consisted of 129 sequences and the multilocus tree consisted of 61 terminals, including sequences from this study (S11 and S12 Tables). Both analyses produced congruent results. *Ca.* M. mahonii was placed within the *Mycoplasma* genus, within a recently characterized lineage of marine taxa, consisting of culturable type strains of *M. marinum* PE$^T$ (isolated from *Octopus vulgaris*) and *M. todarodis* 5H$^T$ (isolated from the squid *Todarodes sagittatus*) [10] based on the multilocus tree. Additionally, 2 marine metagenomes, M. DT_67 and M. DT_68 which were isolated from ocean water sinking particles collected at abyssal depths in the Nort Pacific Subtropical Gyre were also part of the marine clade. The phylogenetic tree for each gene used in the multilocus sequence analysis is shown in the S2–S6 Figs. The 16S rRNA topology placed *Ca.* M. mahonii with these same taxa plus *Candidatus* Mycoplasma corallicola (which was not included in the multilocus tree due to an incomplete genome). Furthermore, the *Ca.* M. mahonii 16S rRNA sequences from both Argentinian waters and North Pacific were ∼99.7% identical and formed a monophyletic clade on the 16S rRNA gene-based phylogenetic tree. The Bayesian analysis also showed

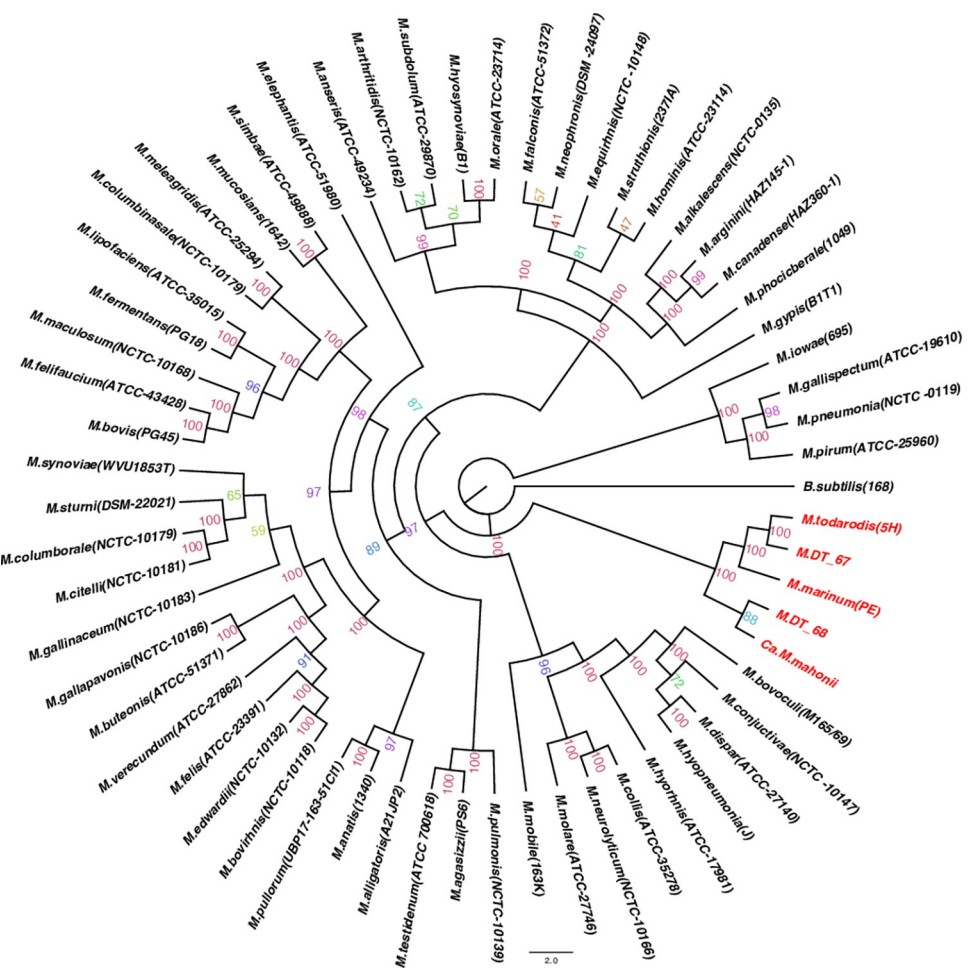

**Fig 5. Multilocus maximum-likelihood phylogenetic tree.** The multilocus phylogenetic tree was generated based on the concatenated sequence of five single-copy housekeeping genes–*recA*, *lepA*, *dnaK*, *ruvB*, and *gmk*. Bootstrap percentage values are shown on the tree. The tree was generated in IQtree with the GTR+F+R6 model. *Ca.* M. mahonii and other sequences making up the distinct marine clade are shaded red.

similar results. *Ca.* M. mahonii formed a monophyletic clade with *Ca.* M. corallicola, M. DT_67, *M. marinum*, and *M. todarodis* in the 16S rRNA-based phylogenetic tree and also formed a monophyletic clade with M. DT_67, M. DT_68, *M. marinum*, and *M. todarodis* in the multilocus tree (S7 and S8 Figs). M. DT_68 lacks a 16S rRNA gene and hence was absent in the 16S rRNA phylogenetic tree.

Additionally, 16S sequence similarity analysis using TrueBac ID from Ezbiocloud [55] indicated that *Ca.* M. mahonii was most similar to *Ca.* M. corallicola (89.41%), *M. todarodis* (89.1%) and M. *marinum* (88.9%). TrueBac ID also indicated high similarity to *M. mobile* (88.9%). However, all the phylogenetic analyses conducted showed M. mobile as an unstable branch as its position varied on the different trees. Consequently, *M. mobile* was not further considered as a closely related species. Genomic features of *Candidatus*. M. mahonii and closely related taxa are shown in Table 1.

ANI similarity values for *Ca.* M. mahonii and its closely related species were 68.2% (*M. marinum*) and 67.4% (*M. todarodis*), while the AAI was 50.3% (*M. marinum*) and 49.25% (*M. todarodis*). A total of 422 orthologous gene groups are shared between *Ca.* M. mahonii, *M.*

**Table 1. Genomic features of *Candidatus* Mycoplasma mahonii and closely related taxa.**

|  | *Candidatus* Mycoplasma mahonii | *M. todarodis* | *M. marinum* |
|---|---|---|---|
| Genome completeness | 97.9% | 85% | 97.1% |
| Genome size (bp) | 796,768 | 1,007,879 | 1,171,149 |
| Number of contigs | 1 | 84 | 128 |
| %GC | 30.1% | 30.95% | 28.41% |
| Number of CDS | 780 | 914 | 1003 |
| Hypothetical genes | 374 | 296 | 328 |
| 16S rRNA gene | 1 | 1 | 1 |
| 23S rRNA gene | 1 | 1 | 1 |
| 5S rRNA gene | 1 | 2 | 2 |
| Number of tRNAs | 31 | 39 | 42 |

*todarodis*, and *M. marinum* (Fig 6). In addition, the dDDH scores from TYGS pairwise comparison of *Ca*. M. mahonii against identified close strains were less than 70% (~21.5% - 27.9%) indicating that *Ca*. M. mahonii is indeed a novel species (S10 Table).

## Description of "*Candidatus* Mycoplasma mahonii"

The category "Candidatus" is used to describe prokaryotic entities for which information other than just a DNA sequence is available but lacks other characteristics required for description according to the International Code of Nomenclature of Bacteria [56]. The *Mycoplasma* genome described here represents a novel species of *Mycoplasma* and is currently the only representative of this candidate species. The species is designated "*Candidatus* Mycoplasma mahonii" (N.L. gen. masc. n. *mahonii*, of Mahon, named in honor of long-time Antarctic collaborator Andrew Mahon). This species was isolated from *Gorgonocephalus chilensis* (collected May 2006, latitude; -54˚49, longitude -60˚16, depth 110m) and assignment to "*Candidatus* Mycoplasma mahonii" is based on (i) the associated 16S rRNA gene sequence; accession number—OP995479), (ii) Similarity index score (ANI) of <95% to closest relatives [7, 10], (iii) 97.9% genome completion (according to CheckM analysis), (iv) primer sequence complementary to a region of 16S rRNA- 5'–ACTCCTACGGGAGGCAGCAGTA–3'.

Genome size was ~796 Kb with a G+C content of 30.1%. Within the genome, single copies of rRNA genes and 31 tRNA genes were identified. KEGG-based analysis identified the presence of the following pathways Embden-Meyerhof-Parnas pathway, F1-type ATP Synthase, Acetate production from acetyl-CoA, Folate (vitamin B9) biosynthesis predicted from 7,8-dihydrofolate, nucleotide sugar biosynthesis pathway amongst others. Additionally, KEGG identified both PTS and ABC transport system genes. The genome sequence can be found under NCBI BioSample ID- SAMN32235174 and Accession ID–CP114583.

## Discussion

A novel *Mycoplasma* species, *Candidatus* Mycoplasma mahonii associated with the basket sea star *Gorgonocephalus chilensis* inhabiting the Argentinian waters, was discovered. This taxon also occurs in *Gorgonocephalus* samples from the North Pacific. Phylogenetically, it is part of a recently characterized clade of non-free-living marine lineage of mollicutes that use marine invertebrate organisms as hosts.

The metagenomic assembly of *Ca*. M. mahonii consists of a single 796,768bp contig with a total of 780 predicted protein-coding sequences (CDS). Genomic features of *Ca*. M. mahonii are comparable to previously described *Mycoplasma* species in several respects: 1) The number

Venn Diagram of Number of Shared Orthologous Genes

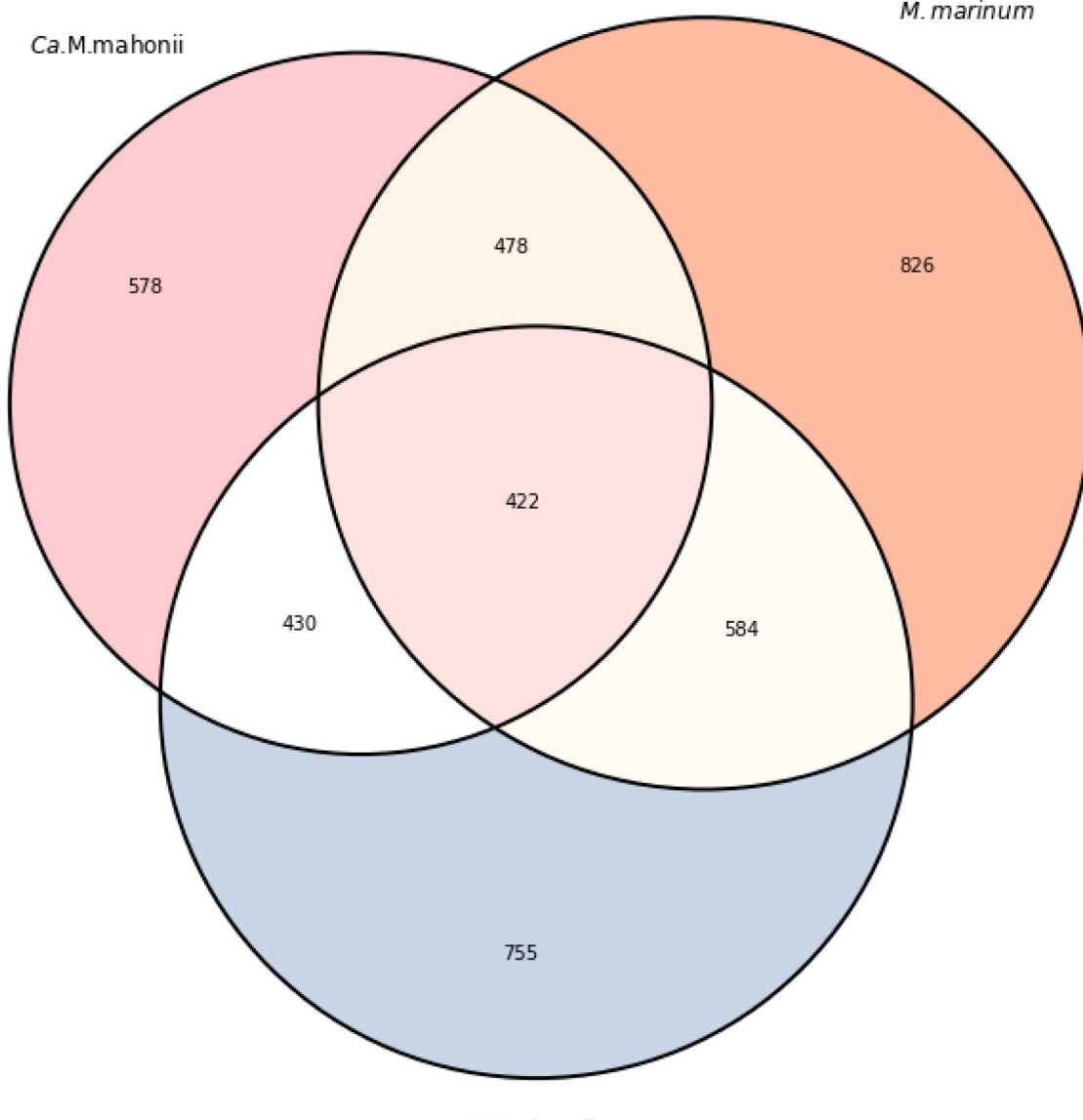

**Fig 6. Venn diagram showing the number of shared orthologous genetic groups (OGs) between *Ca*. M. mahonii, *M. marinum* and *M. todarodis*.**

of predicted CDS in *Ca*. M. mahonii is comparable to other *Mycoplasma* spp., with 635 in *M. mobile* [57], 677 in *M. pneumonia* [58], and 742 in *M. gallisepticum* [51]; 2) The arrangement of rRNA genes is similar to those found in the genome of *M. mobile* [57] and *M. pulmonis* [59]; 3) Although the exact OriC could not be determined, *Ca*. M. mahonii possesses a tandem arrangement of *dnaA* and *dnaN* genes around the OriC as seen in other mycoplasmas [49, 60, 61]; however, in the case of *Ca*. M. mahonii, ribosomal protein L34 (*rmpH* gene) was also located upstream of the *dnaA* gene in an opposite direction; and 4) The Peptide Release Factor (RF-1) which recognizes the stop codons UAA and UAG and terminates translation was

present in *Ca*. M. mahonii, although RF-2 which identifies the UGA stop codon was absent suggesting that this stop codon codes for a protein as seen in other mycoplasmas [62].

Due to its reduced genome size, *Ca*. M. mahonii lacks genes involved in de-novo biosynthesis of nucleotide, lipids, co-factors, and intermediate energy metabolism pathways such as the TCA cycle, citric acid cycle, phosphate pathway, etc., imposing a host-dependent lifestyle on this organism. However, this bacterium is predicted to encode genes involved in the nucleotide salvage pathway such as *nrdA* which converts ribonucleotides to deoxyribonucleotides, and genes involved in nucleotide interconversion such as CTP synthase which converts UTP to CTP as well as permease for pyrimidine and purine transport which allows them to take up these molecules.

Most transport protein-encoding genes in *Ca*. M. mahonii are associated with ABC transport which transports a range of molecules such as peptides, lipids, phosphate, ions, iron, etc., and PTS transport system which transports extracellular sugars such as mannose, fructose, and cellobiose. The presence of these broad substrate transport systems compensates for the lack of various other transport systems such as GLUT (glucose transporters) and may allow the microorganism to obtain nutrients directly from its host rather than synthesizing de-novo, a trend common in mycoplasmas [63]. Additionally, a gene encoding TrkA which is responsible for potassium uptake and is necessary for intracellular survival in prokaryotes was predicted to be present in the genome of *Ca*. M. mahonii.

The lack of a complete TCA cycle, quinones, or cytochromes rules out the possibility of ATP generation through oxidative phosphorylation in this bacterium. Metabolic pathways present in *Ca*. M. mahonii suggest that they are glycolytic species that rely on energy generation through fermentation of sugars, ATP-synthase, pyruvate oxidation to acetate, and hydrolysis of arginine. Additionally, in the non-oxidative phase of the pentose phosphate pathway present in *Ca*. M. mahonii, transaldolase (which catalyzes the transfer of a dihydroxyacetone group from donor compounds to aldehyde acceptor compounds) is absent. This reaction is presumably carried out by an unrecognized protein as the pentose pathway has been reported in other mycoplasmas to be incomplete but functional [57, 64]. In the case of *Ca*. M. mahonii, this reaction is likely carried out by the non-phosphorylating glyceraldehyde-3-phosphate dehydrogenase (GAPN) enzyme, as the gene encoding this enzyme was predicted in the annotation. GAPN reduces NADP to NADPH and can maintain NADPH production in bacteria lacking some pentose phosphate enzyme [65].

The defense systems present in *Ca*. M. mahonii includes R-M Type II system, CRISPR/CAS system, and thioredoxin system. The CRISPR/CAS system and R-M system are a natural pathogenic adaptive immune system that protects prokaryotic organisms against invading nucleic acids most especially viruses.[66, 67]. The R-M system is present in almost all mollicutes sequenced so far [66] while the CRISPR/CAS system has been reported in some but not all mollicutes [68]. On the other hand, the thioredoxin system present in *Ca*. M. mahonii protects it from oxidative stress [69] and has been reported in some *Mycoplasma* species such as *M. suis* [61], *M. bovis* [70], and *M. capricolum* [69].

Virulence factors typically associated with *Mycoplasma* such as adhesion proteins [1], ClpC ATPase [60], variable surface lipoproteins (Vsps) [71, 72], capsular polysaccharides [73], were absent in *Ca*. M. mahonii. Moreover, no virulence factor was detected using the BLAST search tool of the VFDB database, suggesting that *Ca*. M. mahonii is potentially a non-pathogenic *Mycoplasma* species. Interestingly, the absence of virulence factors was also observed in other members of the distinct marine clade of mycoplasmas namely *M. marinum*, *M. todarodis*, and *Ca*. M. corallicola, suggesting that this monophyletic clade of mycoplasmas are commensals and potentially a natural part of its host microbiome.

Lastly, the high percent identities (~99.5%) between the 16S rRNA genes of *Ca*. M. mahonii from *Gorgonocephalus chilensis* found in Argentinian waters and the Northeast Pacific

(*Gorgonocephalus eucnemis*), suggests that this species is broadly distributed and likely native to multiple *Gorgonocephalus* host species. The annotation of *Candidatus* Mycoplasma mahonii, conducted herein, is the first step to understanding the biology and potential pathogenicity of this bacterium. Future studies will expand on this knowledge by focusing on the metabolic pathway interplay between this species and its basket star host.

## Supporting information

**S1 Fig. RAST subsystem feature categories gene counts for *Ca*. M. mahonii, *M. todarodis* and *M. marinum*.**
(TIF)

**S2 Fig. *lepA* gene tree.** The phylogenetic tree was generated in IQtree with the GTR+F+R6 model, bootstrap percentage values are shown on the tree.
(TIF)

**S3 Fig. *gmk* gene tree.** The phylogenetic tree was generated in IQtree with the GTR+F+R5 model, bootstrap percentage values are shown on the tree.
(TIF)

**S4 Fig. *recA* gene tree.** The phylogenetic tree was generated in IQtree with the GTR+F+R5 model, bootstrap percentage values are shown on the tree.
(TIF)

**S5 Fig. *ruvB* gene tree.** The phylogenetic tree was generated in IQtree with the GTR+F+I+G4 model, bootstrap percentage values are shown on the tree.
(TIF)

**S6 Fig. *dnaK* gene tree.** The phylogenetic tree was generated in IQtree with the GTR+F+R5 model, bootstrap percentage values are shown on the tree.
(TIF)

**S7 Fig. Multilocus bayesian phylogenetic tree.** The tree was generated using MrBayes with the GTR+I+G model chosen by JModelTest2. Probability percentage values are shown on the tree. *Ca*. M. mahonii and other sequences making up the distinct marine clade are shaded red.
(TIF)

**S8 Fig. 16S rRNA bayesian phylogenetic tree.** The tree was generated using MrBayes with the GTR+I+G model chosen by JModelTest2. Probability percentage values are shown on the tree. *Ca*. M. mahonii and other sequences making up the distinct marine clade are shaded red. AR– Argentinian waters samples, NP–North Pacific samples.
(TIF)

**S1 Table. Gorgonocephalus samples.**
(XLSX)

**S2 Table. GTDB-TK placement reference.**
(XLSX)

**S3 Table. RAST annotation.**
(XLSX)

**S4 Table. eggNOG annotation.**
(XLSX)

**S5 Table. Genes involved in replication, recombination, and repair.**
(XLSX)

**S6 Table. Genes involved in transcription.**
(XLSX)

**S7 Table. Genes involved in translation, ribosomal structure, and biogenesis.**
(XLSX)

**S8 Table. Genes involved in transport systems.**
(XLSX)

**S9 Table. Genes involved in bacterial secretion system.**
(XLSX)

**S10 Table. dDDH score of pairwise comparison of *Ca*. M. mahonii vs best match type strain genomes.**
(XLSX)

**S11 Table. GenBank IDs of sequences used for multilocus phylogenetic analysis.**
(XLSX)

**S12 Table. Accession numbers of sequences used for 16S rRNA phylogenetic analysis.**
(XLSX)

## Acknowledgments

I would like to thank the members of the Halanych Lab for their valuable information and guidance. Also, Dr. Candace Grimes and Natalie Williams for their helpful insights in this project.

## Author Contributions

**Conceptualization:** Oluchi Aroh, Kenneth M. Halanych.

**Formal analysis:** Oluchi Aroh.

**Funding acquisition:** Kenneth M. Halanych.

**Methodology:** Oluchi Aroh, Mark R. Liles.

**Resources:** Kenneth M. Halanych.

**Visualization:** Oluchi Aroh.

**Writing – original draft:** Oluchi Aroh.

**Writing – review & editing:** Mark R. Liles, Kenneth M. Halanych.

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
