## [Decision Letter · Decision Letter 0]

28 Mar 2023

PONE-D-23-03531Genomic characterization of a novel, widely distributed Mycoplasma Species “Candidatus Mycoplasma mahonii” associated with the brittlestar Gorgonocephalus chilensisPLOS ONE

Dear Dr. Aroh,

Thank you for submitting your manuscript to PLOS ONE. After careful consideration, we feel that it has merit but does not fully meet PLOS ONE’s publication criteria as it currently stands. Therefore, we invite you to submit a revised version of the manuscript that addresses the points raised during the review process.

We look forward to receiving your revised manuscript.

Kind regards,

Rajesh P. Shastry

Academic Editor

PLOS ONE

Journal Requirements:

Reviewers' comments:

Reviewer's Responses to Questions

**Comments to the Author**

1. Is the manuscript technically sound, and do the data support the conclusions?

Reviewer #1: Yes

Reviewer #2: Yes

2. Has the statistical analysis been performed appropriately and rigorously? 

Reviewer #1: Yes

Reviewer #2: Yes

3. Have the authors made all data underlying the findings in their manuscript fully available?

Reviewer #1: Yes

Reviewer #2: Yes

4. Is the manuscript presented in an intelligible fashion and written in standard English?

Reviewer #1: Yes

Reviewer #2: Yes

5. Review Comments to the Author

Reviewer #1: Review of Manuscript: PONE-D-23-03531

“Genomic characterization of a novel, widely distributed Mycoplasma Species “Candidatus Mycoplasma mahonii” associated with the brittlestar Gorgonocephalus chilensis

Author(s): Oluchi, Aroh; Halanych, Kenneth; & Liles, Mark

In this manuscript, Oluchi et al. describe a novel bacterial species belonging to genus Mycoplasma, a significant clade of pathogens causing disease in plants and animals, including humans. The novel organism, here proposed as Candidatus Mycoplasma mahonii, appears to be associated to brittlestars of the genus Gorgoncephalus. In the absence of microscopical evidence, the symbiotic nature of this association is investigated by the authors using their newly obtained meta-genome assembled genome, which suggests an obligatory symbiotic relationship for the bacteria. The virtually complete MAG (CheckM = 97,9%) not only reveals certain reduction of the biosynthetic pathways that would sustain this observation, but also allows the authors to characterize other significant metabolic pathways of this Mycoplasma spp. For instance, systems and pathways involved in replication, secretion or defense, of particular interest in a genus known to cause disease in humans, cattle, fish and shellfish are carefully analyzed. Per contra, the apparent lack of virulence factors, also makes the authors suggest a non-pathogenic lifestyle. Additionally, the MAG is used as well to adopt five single-copy housekeeping genes to produce a multi-locus phylogenomic tree that increases the resolution of the 16S phylogenetic tree constructed. In essence, the phylogeny places Ca. M. mahonii within a monophyletic clade of Mycoplasmas associated exclusively to marine organisms, perhaps the only one in a genus containing over 100 species. Making reference to its biogeography, the authors conclude by suggesting a wide range distribution of this novel bacterial species; this is done based in two different brittlestar species collected from the Southwest Atlantic (Argentina) and Northeast Pacific (Washington, USA).

In my opinion, this is an easy-to-read and sound study that employs appropriate techniques and a wide range of bioinformatic tools and pipelines to investigate a novel bacterial symbiont, which given its phylogenetic position, will certainly assist in the comprehension of marine mycoplasmoses. In fact, among the very few things that I would point out in this beautifully explicative and thoughtful analysis, it is the diluted level of comparison/discussion that makes between the novel and other marine Mycoplasma spp. After all, this is perhaps the principal interest of the paper, as marine Mycoplasmas, whether parasitic or not, are a clear minority and comparatively largely understudied. Furthermore, closely related bacterial lineages are associated with fish and shellfish of commercial relevance that would certainly benefit from equivalent genomic analyses and correlations. In this conceptual line are most of my “major” suggestions. Additionally, I also observed few “minor” specificities that I would need further explained. Also few typos that might be easier to change now than later on, such as those observed in the figures.

Major issues:

Like the authors (lines 265 - 274), I find conflicting the fact that under sequence similarity approaches all Mycoplasma spp., associated to marine animals group together. However, this is not the case under phylogenetic and phylogenomic approaches. It appears particularly problematic to me the presence of M. mobile in a clade formed exclusively by mammal-infecting Mycoplasmas, rather than in the otherwise monophyletic marine clade including M. corallicola, Ca. M. mahonii, M. marinum and M. todarodis. I would be more prone to accept these results if the same existing phylogenetic trees (Figs. 4 & 5) also included the branch support for a Bayesian phylogenetic method. Maybe exploring other maximum likelihood methods and programs could prove interesting and not really time-consuming. Additionally, the branching in the 16S tree might benefit if additional sequences isolated from marine animals (Mycoplasmatales bacteria DT67 and DT68, as well as Oyster Mollicutes MAG) were included, as a wider taxon sampling often helps with small lineages.

Minor issues:

- Since the study involves field research, a brief explanation on the necessity or not of specific permits and approvals is mandatory in the ethics statement.

Abstract:

Lines 21-23 : “yet other than corals, little is known about symbiotic relationships of marine animals that do not live in extreme environments”.

What about parasites of fish and shellfish species with commercial relevance that have been studied in aquaculture facilities for decades? After all, the difficulty to test Koch´s postulates makes, indeed, the study of symbioses in corals particularly complex in comparison to other marine species.

Line 34: Ca. M. Mahonia

Introduction:

Lines 54-56: While I really like this brief but complete review-like phrase, I would find it useful if it specified a bit more regarding the type of “association” (mutualism, parasitism…) as it is relevant for the discussion.

Line 65: “first isolation” or first genetic isolation?

Materials & Methods:

Line 74: “DNA was isolated and sequenced using PacBio…” How it was isolated? Is it the same as in lines 107 and 108 later on? How was the library preparation conducted for the PacBio CCS?

Line 83: As it has been isolated from the metagenome that constitutes the Gorgoncephalus brittlestar, I think it would be more appropriate to use the concept MAG rather than genome.

Line 122: I found that the use of “assembled” in this section might be a bit confusing, if what it was done, was in fact pairing bidirectional reads for additional sequence robustness.

Lines 126-128: “to overcome the bias caused by single-gene phylogeny…” What type of bias would it be? I find “a higher resolution of the phylogenetic relationship of this species” explanatory enough to also include “robust representation of genomic data” as an additional point within the phrase.

Line 134: Maximum likelihood analysis with bootstrap support?

Lines 138 and 147: Was any method used to identify the best substitution model? Which? Why are they different GTR+F+R10 and GTR+F+R6 if not?

Line 142: DNA sequence(s)

Line 143: Is there any particular reason for not using Mafft, as before?

Results:

Lines 157-159: If the three contigs that were identified to belong to Mycoplasma sp. are available, why not to contrast wether they belong to the same organisms rather than writing “hypothesized”? Do they have exactly the same 16S rRNA?

Line 164: “Robustly” – Is there any stats available?

Line 265: Which sequence similarity approaches?

Discussion:

Lines 355-357: If the description of the lifestyle (mutualistic, parasitic, …) of related Mycoplasma species was a little bit more developed in the introduction and discussion, the apparent absence of virulence factors would be very interesting, specially if absent in closely related species or in the marine-borne monophyletic clade. Both from an evolutionary perspective and from its ecological implications.

Lines 360-361: Is it not worthy to check if other assembled metagenomes of Gorgoncephalus species worlwide are associated to Ca M. mahonii as well? It could certainly provide more robustness to the actual status of “suggestion”. After all, its wide-range distribution is stated and not suggested in the title of the paper.

Figures:

Figure 2: While certainly interesting, an equivalent figure comparing Ca. M. mahonii with its closer relatives or with better studied Mycoplasma spp. could definitely assist to contextualize the characterized MAG.

Figure 4: M. coracoilla?

The phylogenetic tree could include Mycoplasmatales bacteria DT67 and DT68, as well as Oyster mollicutes MAG, which is associated to Crassostrea sp. It would be interesting to see if a wider representation of Mycosplasma spp. associated to marine animals makes the clade monophyletic or not, as the presence of M. mobile in a clade with mammal-infecting Mycoplasma spp. (Fig. 5) is, as the authors signaled (Lines 265-270) conflicting.

Reviewer #2: The manuscript by Arch et al. describe the genome of a novel symbiotic Mycoplasma species derived from Gorgonocephalus chilensis. The genome (796Kb) showed completeness. The manuscript is focused on the genome analysis and conclusion/metabolic features drawn from the genome annotation as physiological experiments were not (could not be) performed. In subsequent experiments the distribution of the new species “Candidatus Mycoplasma mahonii” ´specifically associated with marine animals was analyzed based o on 16S rRNA PCR-screening. The authors detected Ca. M. mahonii in Gorgonocephalus eucnemis from the Northwest Pacific and other Gorgonocephalus chilensis from Argentinian waters. The genome-focused analysis is sound and straight-forward and in most aspects state of the art.

I have only a few specific Comments:

Line 65 in the Introduction "..... first isolation and characterization...". This line is misleading as it implies that an isolated "living" organism was at hand, but to my understanding of the paper this was not the case.

Lines 81-87. The GTDB-TK analysis and the placement in the GTDB reference should be presented at least as supplementary materials.

As GTDB was already mentioned, why was not the now accepted whole-genome taxonomy used for the phylogenetic placement, which is more reliable for placement than MLST or 16S. The GGTDB-Tk could be used for whole-genome-based taxonomy with the Type (Strain) Genome Server (TYGS). The ANIm method, which is provided by pyani, should be employed for in-depth phylogenetic analysis.

6. PLOS authors have the option to publish the peer review history of their article (what does this mean?). If published, this will include your full peer review and any attached files.

Reviewer #1: No

Reviewer #2: **Yes: **Rolf Daniel

---

## [Author Response · Author response to Decision Letter 0]

22 May 2023

Dear Dr. Shastry, 

We once again appreciate the efforts and time that you and the reviewers have put into reviewing this manuscript. We have acted on the helpful suggestions and provided a point-by-point based response. Responses are given below the appropriate suggestion. 

5. Review Comments to the Author

Reviewer #1: Review of Manuscript: PONE-D-23-03531

"Genomic characterization of a novel, widely distributed Mycoplasma Species "Candidatus Mycoplasma mahonii" associated with the brittlestar Gorgonocephalus chilensis

Author(s): Oluchi, Aroh; Halanych, Kenneth; & Liles, Mark

In this manuscript, Oluchi et al. describe a novel bacterial species belonging to genus Mycoplasma, a significant clade of pathogens causing disease in plants and animals, including humans. The novel organism, here proposed as Candidatus Mycoplasma mahonii, appears to be associated to brittlestars of the genus Gorgoncephalus. In the absence of microscopical evidence, the symbiotic nature of this association is investigated by the authors using their newly obtained meta-genome assembled genome, which suggests an obligatory symbiotic relationship for the bacteria. The virtually complete MAG (CheckM = 97,9%) not only reveals certain reduction of the biosynthetic pathways that would sustain this observation, but also allows the authors to characterize other significant metabolic pathways of this Mycoplasma spp. For instance, systems and pathways involved in replication, secretion or defense, of particular interest in a genus known to cause disease in humans, cattle, fish and shellfish are carefully analyzed. Per contra, the apparent lack of virulence factors, also makes the authors suggest a non-pathogenic lifestyle. Additionally, the MAG is used as well to adopt five single-copy housekeeping genes to produce a multi-locus phylogenomic tree that increases the resolution of the 16S phylogenetic tree constructed. In essence, the phylogeny places Ca. M. mahonii within a monophyletic clade of Mycoplasmas associated exclusively to marine organisms, perhaps the only one in a genus containing over 100 species. Making reference to its biogeography, the authors conclude by suggesting a wide range distribution of this novel bacterial species; this is done based in two different brittlestar species collected from the Southwest Atlantic (Argentina) and Northeast Pacific (Washington, USA).

In my opinion, this is an easy-to-read and sound study that employs appropriate techniques and a wide range of bioinformatic tools and pipelines to investigate a novel bacterial symbiont, which given its phylogenetic position, will certainly assist in the comprehension of marine mycoplasmoses. In fact, among the very few things that I would point out in this beautifully explicative and thoughtful analysis, it is the diluted level of comparison/discussion that makes between the novel and other marine Mycoplasma spp. After all, this is perhaps the principal interest of the paper, as marine Mycoplasmas, whether parasitic or not, are a clear minority and comparatively largely understudied. Furthermore, closely related bacterial lineages are associated with fish and shellfish of commercial relevance that would certainly benefit from equivalent genomic analyses and correlations. In this conceptual line are most of my "major" suggestions. Additionally, I also observed few "minor" specificities that I would need further explained. Also few typos that might be easier to change now than later on, such as those observed in the figures.

We are grateful to Reviewer 1 for the kind words about the manuscript. We have added more comparison to other marine Mycoplasma spp. in our discussion.

Major issues:

Like the authors (lines 265 - 274), I find conflicting the fact that under sequence similarity approaches all Mycoplasma spp., associated to marine animals group together. However, this is not the case under phylogenetic and phylogenomic approaches. It appears particularly problematic to me the presence of M. mobile in a clade formed exclusively by mammal-infecting Mycoplasmas, rather than in the otherwise monophyletic marine clade including M. corallicola, Ca. M. mahonii, M. marinum and M. todarodis. I would be more prone to accept these results if the same existing phylogenetic trees (Figs. 4 & 5) also included the branch support for a Bayesian phylogenetic method. Maybe exploring other maximum likelihood methods and programs could prove interesting and not really time-consuming. Additionally, the branching in the 16S tree might benefit if additional sequences isolated from marine animals (Mycoplasmatales bacteria DT67 and DT68, as well as Oyster Mollicutes MAG) were included, as a wider taxon sampling often helps with small lineages.

Response: We thank the reviewer for raising this concern. We have included the sequences mentioned above to our phylogenetic analyses that now include a Bayesian inference via MrBayes as well as RAxML likelihood run. M. DT_68 has only 5S rRNA hence was not included in the 16S tree while the Oyster MAGs were only present in the 16S tree. RAxML and IQ tree gave similar results, hence, to remove the redundancy of using the same methods, (IQ tree employs maximum likelihood), phylogenetic trees from RaxML were not added to the supplementary materials. 

We used EzBiocloud for our sequence similarity analysis, hence the result is restricted to Mycoplasma species present in the EzBiocloud database. Therefore, the similarity and phylogenetic analysis difference isn’t surprising as the phylogenetic trees are more robust (have more sequences and compared more than 1 gene). Issues with sequence similarity approaches for inferring relationships are well known, including the inability to account for multiple substitutions and lack of statistical support, hence, phylogenetic analysis is the preferred method of inference for evolutionary relationships. We reviewed our Results and Discussion sections of the manuscript to make sure it is clear that conclusions about relatedness were not based on sequence similarity analysis alone.

In reference specifically to M. mobile, this organism was previously described as belonging to the “M. hominis group of Mycoplasma” (Jaffe et al. 2004). Additionally, its position was inconsistent through all the inferred phylogenetic trees, this instability has also been noted in previous work, specifically, the (Ramirez et al. 2019) study that characterized M. marinum and M. todarodis. Interestingly, this Mycoplasma occurs on fish that lives in fresh to brackish waters whereas the marine Mycoplasma herein occurs on invertebrates. Given the difference in vertebrate physiology and especially immunity, this could explain the placement of M. mobile. We have revised the text to make the habitat differences clear.

Minor issues:

- Since the study involves field research, a brief explanation on the necessity or not of specific permits and approvals is mandatory in the ethics statement.

Response: The manuscript has been updated.

Abstract:

Lines 21-23 : "yet other than corals, little is known about symbiotic relationships of marine animals that do not live in extreme environments".

What about parasites of fish and shellfish species with commercial relevance that have been studied in aquaculture facilities for decades? After all, the difficulty to test Koch´s postulates makes, indeed, the study of symbioses in corals particularly complex in comparison to other marine species.

Response: The manuscript has been revised to better convey accuracy.

Line 34: Ca. M. Mahonia

Response: Corrected.

Introduction:

Lines 54-56: While I really like this brief but complete review-like phrase, I would find it useful if it specified a bit more regarding the type of "association" (mutualism, parasitism...) as it is relevant for the discussion.

Response:The manuscript has been revised.

Line 65: "first isolation" or first genetic isolation?

Response: Corrected.

Materials & Methods:

Line 74: "DNA was isolated and sequenced using PacBio..." How it was isolated? Is it the same as in lines 107 and 108 later on? How was the library preparation conducted for the PacBio CCS?

Response: The manuscript has been updated.

Line 83: As it has been isolated from the metagenome that constitutes the Gorgoncephalus brittlestar, I think it would be more appropriate to use the concept MAG rather than genome.

Response: The manuscript has been revised.

Line 122: I found that the use of "assembled" in this section might be a bit confusing, if what it was done, was in fact pairing bidirectional reads for additional sequence robustness.

Response: The manuscript has been revised to better convey its intended meaning.

Lines 126-128: "to overcome the bias caused by single-gene phylogeny..." What type of bias would it be? I find "a higher resolution of the phylogenetic relationship of this species" explanatory enough to also include "robust representation of genomic data" as an additional point within the phrase.

Response: Corrected.

Line 134: Maximum likelihood analysis with bootstrap support?

Response: Corrected.

Lines 138 and 147: Was any method used to identify the best substitution model? Which? Why are they different GTR+F+R10 and GTR+F+R6 if not?

Response: IQtree employs ModelFinder tool to find the best substitution model for each phylogenetic tree. The manuscript has been updated to reflect this.

Line 142: DNA sequence(s)

Response: The manuscript has been corrected.

Line 143: Is there any particular reason for not using Mafft, as before?

Response: The analyses and manuscript have been updated to use Mafft for all alignment to provide consistency.

Results:

Lines 157-159: If the three contigs that were identified to belong to Mycoplasma sp. are available, why not to contrast wether they belong to the same organisms rather than writing "hypothesized"? Do they have exactly the same 16S rRNA?

Response: No they do not have the exact same 16S rDNA. The manuscript has been revised to reflect this.

Line 164: "Robustly" – Is there any stats available?

Response: Yes, information added.

Line 265: Which sequence similarity approaches?

Response: The manuscript has been revised to better inform the reader. 

Discussion:

Lines 355-357: If the description of the lifestyle (mutualistic, parasitic, ...) of related Mycoplasma species was a little bit more developed in the introduction and discussion, the apparent absence of virulence factors would be very interesting, specially if absent in closely related species or in the marine-borne monophyletic clade. Both from an evolutionary perspective and from its ecological implications.

Response: The manuscript has been revised

Lines 360-361: Is it not worthy to check if other assembled metagenomes of Gorgoncephalus species worlwide are associated to Ca M. mahonii as well? It could certainly provide more robustness to the actual status of "suggestion". After all, its wide-range distribution is stated and not suggested in the title of the paper.

Response: There are no other assembled metagenomes of Gorgonocephalus species

Figures:

Figure 2: While certainly interesting, an equivalent figure comparing Ca. M. mahonii with its closer relatives or with better studied Mycoplasma spp. could definitely assist to contextualize the characterized MAG.

Response: The figure has been added to the supplementary file

Figure 4: M. coracoilla?

The phylogenetic tree could include Mycoplasmatales bacteria DT67 and DT68, as well as Oyster mollicutes MAG, which is associated to Crassostrea sp. It would be interesting to see if a wider representation of Mycosplasma spp. associated to marine animals makes the clade monophyletic or not, as the presence of M. mobile in a clade with mammal-infecting Mycoplasma spp. (Fig. 5) is, as the authors signaled (Lines 265-270) conflicting.

Response: The addition of other Mycoplasma species resulted in a monophyletic marine clade, consistent with previous results. Additionally, M. mobile was unstable as its position varied across all the phylogenetic trees. This inconsistency could be because the M. mobile sequence available from GenBank database requires further verification or because the Mycoplasma in the marine clade described here infect only invertebrate host, hence is evolutionally different from M. mobile which infects a mainly freshwater vertebrate (fish).

Reviewer #2: The manuscript by Arch et al. describe the genome of a novel symbiotic Mycoplasma species derived from Gorgonocephalus chilensis. The genome (796Kb) showed completeness. The manuscript is focused on the genome analysis and conclusion/metabolic features drawn from the genome annotation as physiological experiments were not (could not be) performed. In subsequent experiments the distribution of the new species "Candidatus Mycoplasma mahonii" ´specifically associated with marine animals was analyzed based o on 16S rRNA PCR-screening. The authors detected Ca. M. mahonii in Gorgonocephalus eucnemis from the Northwest Pacific and other Gorgonocephalus chilensis from Argentinian waters. The genome-focused analysis is sound and straight-forward and in most aspects state of the art.

I have only a few specific Comments:

Line 65 in the Introduction "..... first isolation and characterization...". This line is misleading as it implies that an isolated "living" organism was at hand, but to my understanding of the paper this was not the case.

Response: Corrected.

Lines 81-87. The GTDB-TK analysis and the placement in the GTDB reference should be presented at least as supplementary materials.

Response: The information has been added to the supplementary material.

As GTDB was already mentioned, why was not the now accepted whole-genome taxonomy used for the phylogenetic placement, which is more reliable for placement than MLST or 16S. The GGTDB-Tk could be used for whole-genome-based taxonomy with the Type (Strain) Genome Server (TYGS). The ANIm method, which is provided by pyani, should be employed for in-depth phylogenetic analysis.

Response: In this case, GTBD-tk was used to place this organism generally, and phylogenetic analyses were used to more precisely and appropriately pin down the placement of this organism. There is agreement in the approaches. More specifically, GTBD uses a number of factors to place taxa including average nucleotide identity and relative divergence. These essentially similarity-based metrics are not appropriate for placing taxa phylogenetically. Moreover, the parsimony method employed by TYGS for phylogeny is problematic by modern standards. It is also dependent on reference genomes that can have more sampling limitations. However, digital DNA: DNA hybridization scores were calculated using TYGS (See supplementary files). 

Additionally, The Ezbiocloud ANI calculator has already been used to calculate ANI scores to determine whole-genome similarity (which is what ANIm by pyani does too), hence, to avoid redundancy, the pyani ANIm calculator was not employed.

---

## [Decision Letter · Decision Letter 1]

15 Jun 2023

PONE-D-23-03531R1Genomic characterization of a novel, widely distributed Mycoplasma Species “Candidatus Mycoplasma mahonii” associated with the brittlestar Gorgonocephalus chilensisPLOS ONE

Dear Dr. Aroh,

Thank you for submitting your manuscript to PLOS ONE. After careful consideration, we feel that it has merit but does not fully meet PLOS ONE’s publication criteria as it currently stands. Therefore, we invite you to submit a revised version of the manuscript that addresses the points raised during the review process.

We look forward to receiving your revised manuscript.

Kind regards,

Rajesh P. Shastry

Academic Editor

PLOS ONE

Journal Requirements:

Reviewers' comments:

Reviewer's Responses to Questions

**Comments to the Author**

1. If the authors have adequately addressed your comments raised in a previous round of review and you feel that this manuscript is now acceptable for publication, you may indicate that here to bypass the “Comments to the Author” section, enter your conflict of interest statement in the “Confidential to Editor” section, and submit your "Accept" recommendation.

Reviewer #1: All comments have been addressed

Reviewer #3: All comments have been addressed

2. Is the manuscript technically sound, and do the data support the conclusions?

Reviewer #1: Yes

Reviewer #3: Partly

3. Has the statistical analysis been performed appropriately and rigorously? 

Reviewer #1: Yes

Reviewer #3: N/A

4. Have the authors made all data underlying the findings in their manuscript fully available?

Reviewer #1: Yes

Reviewer #3: No

5. Is the manuscript presented in an intelligible fashion and written in standard English?

Reviewer #1: Yes

Reviewer #3: Yes

6. Review Comments to the Author

Reviewer #1: Review of Manuscript: PONE-D-23-03531R1

“Genomic characterization of a novel, widely distributed Mycoplasma Species “Candidatus Mycoplasma mahonii” associated with the brittlestar Gorgonocephalus chilensis

Author(s): Oluchi, Aroh; Halanych, Kenneth; & Liles, Mark

In this reviewed manuscript, the authors have done substantial work to adapt/explain the reviewers’ concerns, particularly those referring to the phylogenetic position of the novel Mycoplasma species and its related taxa. My opinion is that the manuscript has benefited from not-demanded structural changes as well, particularly in the introduction and discussion. I also appreciated a more clear description of the symbiotic relationships between the different Mycoplasma spp. and their hosts. Finally, I have also noticed a higher level of contingency when speaking about the distribution of the species, which I consider positive with the data at hand. Based on the above, I have no further major recommendations for the authors rather than some minor changes which will be most likely corrected in later steps if the manuscript is eventually accepted.

For instance, lines 52- 53: “Pathogenic Mycoplasmas are responsible for numerous respiratory and other infections including pneumonia”. Is not pneumonia a respiratory infection? I would recommend rewording. Or line 60: Salmonid(s)?: Line 123: “search BLAST search tool”; & Line 153: “The second analyses”? To pinpoint few examples.

Reviewer #3: Candidatus or Ca. should be in italics. Mycolasma mahoni should not be italicized. Please maintain uniformity.

In addition, please check all scientific names for italics.

Size and accession number for 16S rRNA genes (Ca. M. mahoni and reference species) used in the study need to be specified. Type species of Mycoplasma need to be shown in the phylogenetic tree

Improvise the figure quality. Resolution of image is poor. Fig. 4 and Fig. 5 are not readable.

Please correct the erroneous genome size given in Fig. 1 (796786 kb).

Genome of Ca. M. Mahoni is not available in the NCBI database for review.

Closely related Culturable type strain(s) (if exist) of Mycoplasma under investigation need to be emphasized.

7. PLOS authors have the option to publish the peer review history of their article (what does this mean?). If published, this will include your full peer review and any attached files.

Reviewer #1: No

Reviewer #3: No

---

## [Author Response · Author response to Decision Letter 1]

1 Aug 2023

Dear Dr. Shastry, 

Thank you for the recent decision letter. Comments/responses are given below the comments in blue Calibri font

Reviewer #1: Review of Manuscript: PONE-D-23-03531R1

“Genomic characterization of a novel, widely distributed Mycoplasma Species “Candidatus Mycoplasma mahonii” associated with the brittlestar Gorgonocephalus chilensis

Author(s): Oluchi, Aroh; Halanych, Kenneth; & Liles, Mark

In this reviewed manuscript, the authors have done substantial work to adapt/explain the reviewers’ concerns, particularly those referring to the phylogenetic position of the novel Mycoplasma species and its related taxa. My opinion is that the manuscript has benefited from not-demanded structural changes as well, particularly in the introduction and discussion. I also appreciated a more clear description of the symbiotic relationships between the different Mycoplasma spp. and their hosts. Finally, I have also noticed a higher level of contingency when speaking about the distribution of the species, which I consider positive with the data at hand. Based on the above, I have no further major recommendations for the authors rather than some minor changes which will be most likely corrected in later steps if the manuscript is eventually accepted.

For instance, lines 52- 53: “Pathogenic Mycoplasmas are responsible for numerous respiratory and other infections including pneumonia”. Is not pneumonia a respiratory infection? I would recommend rewording. Or line 60: Salmonid(s)?: Line 123: “search BLAST search tool”; & Line 153: “The second analyses”? To pinpoint few examples. Reviewer #3: Candidatus or Ca. should be in italics. Mycolasma mahoni should not be italicized. Please maintain uniformity. In addition, please check all scientific names for italics

Manuscript has been revised and corrected.

Size and accession number for 16S rRNA genes (Ca. M. mahoni and reference species) used in the study need to be specified. 

Information has been added to the supplementary file.

Type species of Mycoplasma need to be shown in the phylogenetic tree

Improvise the figure quality. Resolution of image is poor. Fig. 4 and Fig. 5 are not readable.

Please correct the erroneous genome size given in Fig. 1 (796786 kb).

Figures have been revised accordingly.

Genome of Ca. M. Mahoni is not available in the NCBI database for review.

The genomes are now available under the accessions listed in the manuscript.

Closely related Culturable type strain(s) (if exist) of Mycoplasma under investigation need to be emphasized.

We already mentioned the closely related strains M. marinum and M. todarodis, however, we have modified the wording to make it clear these are closely related culturable type strains.

---

## [Editor Report · Decision Letter 2]

4 Aug 2023

Genomic characterization of a novel, widely distributed Mycoplasma Species “Candidatus Mycoplasma mahonii” associated with the brittlestar Gorgonocephalus chilensis

PONE-D-23-03531R2

Dear Dr. Aroh,

We’re pleased to inform you that your manuscript has been judged scientifically suitable for publication and will be formally accepted for publication once it meets all outstanding technical requirements.

Kind regards,

Rajesh P. Shastry

Academic Editor

PLOS ONE
---

## [Editor Report · Acceptance letter]

14 Aug 2023

PONE-D-23-03531R2 

Genomic characterization of a novel, widely distributed *Mycoplasma* Species *“Candidatus* Mycoplasma mahonii” associated with the brittlestar *Gorgonocephalus chilensis*

Dear Dr. Aroh:

I'm pleased to inform you that your manuscript has been deemed suitable for publication in PLOS ONE. Congratulations! Your manuscript is now with our production department. 

Kind regards, 

on behalf of

Dr. Rajesh P. Shastry 

Academic Editor

PLOS ONE